# Effect of perioperative dexmedetomidine on sleep quality in adult patients after noncardiac surgery: A systematic review and meta-analysis of randomized trials

Lin Wang[1,2,3], Xin-Quan Liang[1], Yan-Xia Sun[2,3], Zhen Hua[2,3]*, Dong-Xin Wang[1,4]*

1 Department of Anesthesiology, Peking University First Hospital, Beijing, China, 2 Department of Anesthesiology, Beijing Hospital, National Center of Gerontology, Beijing, China, 3 Institute of Geriatric Medicine, Chinese Academy of Medical Sciences, Beijing, China, 4 Outcomes Research Consortium, Houston, Texas, United States of America

* hua1013@163.com (ZH); dxwang65@bjmu.edu.cn, wangdongxin@hotmail.com (DXW)

**Data Availability Statement:** All relevant data are within the manuscript and its Supporting information files.

## Abstract

### Background

Dexmedetomidine may improve sleep quality after surgery, but conflicting results also exist. Herein, we explored the effects of perioperative dexmedetomidine on postoperative sleep quality in adult patients.

### Methods

In this systematic review and meta-analysis, randomized controlled trials investigating the effects of perioperative dexmedetomidine on sleep quality after noncardiac surgery were retrieved from Cochrane Library, PubMed, and EMBASE from inception to January 12, 2023, and updated on March 15, 2024. The Cochrane Collaboration's tool was applied to assess risk of bias. A random-effects model was used for meta-analysis. The primary outcome was the subjective sleep quality score on the first night after surgery.

### Results

A total of 29 trials containing 5610 participants were included. The subjective sleep score on the first postoperative night was lower (better) with dexmedetomidine than with placebo (SMD [standardized mean difference] = -0.8, 95% CI -1.1 to -0.6, p<0.00001; $I^2$ = 93%; 22 trials; n = 4611). Sensitivity analysis showed that overall conclusion was not changed (SMD = -0.8, 95% CI -1.1 to -0.5, p<0.00001; $I^2$ = 93%; 14 trials; n = 3846). Results of polysomnographic monitoring showed improved sleep structure with dexmedetomidine on the first night after surgery, as manifested by increased sleep efficiency index and stage N2 sleep and decreased arousal index and stage N1 sleep.

### Conclusions

This systematic review suggests that, among patients who underwent noncardiac surgery, perioperative dexmedetomidine administration may improve early postoperative sleep

**Funding:** Funded by National Natural Science Foundation of China (Major Program No. 82293644) and National High Level Hospital Clinical Research Funding (High Quality Clinical Research Project of Peking University First Hospital No. 2022CR78). The funders had no role in study design, data collection and analysis, decision to publish, or preparation of the manuscript.

**Competing interests:** The authors have declared that no competing interests exist.

quality pattern. However, the resulting evidence were of low or very low qualities and further studies are required to confirm our results.

## PROSPERO registration number

CRD42023390972.

## 1. Introduction

Sleep is a periodic biological behavior. Normal sleep is extremely important for timely clearance of brain metabolites and maintenance of brain function [1]. Persistent sleep disturbances may lead to continuous activation of inflammation and produce a series of adverse consequences [2]. Due to acute surgical pain, inflammatory response, catheter-related discomfort, nursing activities, and environmental interference, postoperative sleep disturbances are common [3] and usually manifested as shortened total sleep time, lowered sleep efficiency [4], and disordered sleep structure at night [5, 6]. The decline of sleep quality after surgery is also associated with a variety of adverse outcomes, including increased pain sensitivity [7, 8], delirium occurrence [9, 10], and adverse cardiovascular events [11], and reduced long-term quality of life [12].

Dexmedetomidine is a highly selective α2-adrenergic receptor agonist. It exerts sedative and hypnotic effects by activating the endogenous sleep-promoting pathways and results in a state like non-rapid eye movement sleep with little effect on spontaneous respiration [13]. Studies showed that night-time infusion of low-dose dexmedetomidine improves sleep structure and quality and reduces delirium in intensive care unit (ICU) patients after surgery [14, 15]. However, conflicting results also exist. For example, a trial reported that patients given dexmedetomidine during mechanical ventilation expressed more discomfort and sleep difficulty when compared with propofol [16]. In another trial of patients undergoing transurethral resection of the prostate, those who received dexmedetomidine sedation had worse sleep parameters after surgery [17]. The effect of dexmedetomidine on postoperative sleep quality remains controversial.

In recent years, more evidence emerged regarding the effect of dexmedetomidine on sleep quality, but usually with limited sample size and variable quality. An early systematic review included both randomized trials and observational studies and explored the effect of dexmedetomidine on postoperative sleep quality; however, only qualitative analysis was done without data extraction, synthesis, and analysis [18]. A 2023 systematic review and meta-analysis included 5 randomized trials with 381 participants and investigated the effect of dexmedetomidine on postoperative sleep structure as assessed with polysomnography; but the effect on subjective sleep quality remains largely unknown [19]. We therefore conducted this systematic review and meta-analysis of randomized trials to investigate the effect of perioperative dexmedetomidine on both subjective and objective sleep quality in adult patients after surgery.

## 2. Methods

### 2.1 Protocol and registration

This systematic review and meta-analysis were conducted in accordance with the Preferred Reporting Items for Systematic Review and Meta-Analysis (PRISMA) guidelines (S1 File). The protocol was registered with the PROSPERO database (CRD42023390972). Our predefined

primary outcome was the subjective sleep quality score on the first night after surgery. We added exploratory and safety outcomes after data collection because these results were important to clarify the effects of dexmedetomidine.

## 2.2 Inclusion and exclusion criteria

The inclusion criteria were randomized trials that (1) were conducted in adult patients (aged 18 years or older) who underwent noncardiac surgery, (2) compared perioperative dexmedetomidine versus control, and (3) reported subjective sleep quality and/or sleep structure parameters as outcomes. The exclusion criteria were studies that (1) investigated the effect of non-intravenous dexmedetomidine, (2) did not include a placebo group (0.9% saline), or (3) did not have full-text paper published in peer-reviewed journals in English.

## 2.3 Search strategy

We searched the databases of Cochrane Library, PubMed, and EMBASE to retrieve studies investigating the effects of dexmedetomidine on postoperative sleep quality from inception to January 12, 2023; this search was updated on March 15, 2024. Medical Subject Headings (MeSH) and free words were combined to find potential articles. We also searched the references of the included articles and the ClinicalTrial.gov. The search strategies for each database are provided in S2 File.

## 2.4 Study selection and data extraction

Two reviewers (LW and X-QL) screened the literature independently at the same time. After eliminating the duplicates, potential eligible articles were screened firstly by reading the titles and abstracts, and then by reading the full texts. Only studies that met the inclusion and exclusion criteria were selected for this systematic review and meta-analysis. Disagreements were resolved through discussion or referred to a senior reviewer (ZL) for judgment.

Data extraction was performed by two reviewers (LW and X-QL) simultaneously and independently. The following data were extracted: (1) basic information of the included studies: first author, year of publication, and sample size; (2) basic characteristics of the subjects: age, sex, type of anesthesia, type of surgery, and postoperative analgesia; (3) strategy and timing of interventions; (4) main outcome parameters of interest; and (5) key elements for risk of bias assessment. Disagreements were resolved through discussion or referred to the senior reviewer (ZL) for judgment. The corresponding authors were contacted for missing data or un-reported information. If there were no responses after three contacts, we estimated data according to figures in the original articles.

## 2.5 Risk of bias assessment

The risks of bias of included studies were assessed using the Cochrane Collaboration's Risk of Bias tool 2 [20]. Specifically, six domains were evaluated and included randomization process, deviations from intended interventions, missing outcome data, measurement of the outcome, selection of the reported result, and overall biases. Two reviewers (LW and X-QL) independently evaluated the risk of bias, cross-checked the results, and resolved any disagreements by discussion.

## 2.6 Statistical analysis

Our primary outcome was the score of subjective sleep quality on the first night after surgery. Exploratory outcomes included polysomnographic (PSG) parameters on the first night after

surgery, such as sleep efficiency index (SEI, %), arousal index (AI, times/h), and percentages of rapid eye movement (%REM) sleep and stage 1 (%N1), stage 2 (%N2), and stage 3 (%N3) of non-REM sleep, as well as subjective sleep quality scores on the second and third nights and at 1 week or later after surgery. Numeric rating scale of pain at 24 hours, morphine equivalent consumption within 7 days, and delirium incidence after surgery were also analyzed. Safety outcomes included the incidences of bradycardia and hypotension.

For subject sleep scores, results that were assessed with a 0–100 scale were converted according to a 0–10 scale, with higher score indicating worse sleep quality; results assessed with other tools were converted to the same direction, with higher score indicating worse sleep quality. We then calculated standardized mean difference (SMD) and 95% CI according to the Cochrane Handbook [21]. For other continuous data, mean difference (MD) and 95% CI were calculated. For dichotomous data, risk ratio (RR) and 95% confidence interval (CI) were calculated [21].

When included studies reported only median and interquartile range, we contacted the authors to obtain the corresponding mean and standard deviation (SD). If there was no response after three contacts, mean and SD were estimated according to the reported formula [21]. When included studies contained two dexmedetomidine groups with different doses, we pooled mean and SD according to the reported formula [21].

The heterogeneity of the included studies was evaluated using the Cochran chi-square test (Q) and $I^2$ statistic. Considering the clinical heterogeneity of the included studies, random-effects models were used for meta-analyses. For the primary outcome, subgroup analyses were performed according to patients' age ($\geq$65 years or <65 years) and timing of dexmedetomidine administration (intraoperative, postoperative, or both). As sensitivity analyses, meta-analyses were reconducted after excluding selected studies that had small sample sizes (less than 200 participants) and reported medians as efficacy outcomes [21] or were judged at high risk of bias. Publication bias of the primary outcome was determined by the funnel plot and Egger test. A modified Grading of Recommendations, Assessment, Development, and Evaluation (GRADE) approach was used to evaluate the quality of evidence [22]. A p value of <0.05 was considered statistically significant. Statistical analysis was performed using the Review Manager (version 5.4., The Cochrane Collaboration, 2020).

## 3. Results

### 3.1 Results of literature search and screening

A total of 1387 articles were retrieved according to the search strategy. Two hundred sixty-four articles were excluded for duplication. After screening the titles and abstracts, 1029 articles did not meet the inclusion/exclusion criteria and were excluded. The full texts of the remaining 94 articles were read and re-assessed for eligibility, and finally a total of 29 articles that met the inclusion/exclusion criteria were included for quantitative analysis (Fig 1; S1 Table) [14, 15, 17, 23–48].

### 3.2 Characteristics of the included studies

Characteristics of the included studies were published between 2015 and 2024 and are listed in Table 1. A total of 29 trials with 5610 patients were included, with 2868 patients in the dexmedetomidine group (dexmedetomidine administered intravenously during the perioperative period) and 2742 patients in the placebo group (normal saline administered intravenously at the same rate and volume for the same duration).

The average age of participants was <65 years in 16 of the included studies [23–25, 28, 29, 31, 35, 36, 38, 39, 42–47] and was $\geq$65 years in 13 others [14, 15, 17, 26, 27, 30, 32–34, 37, 40,

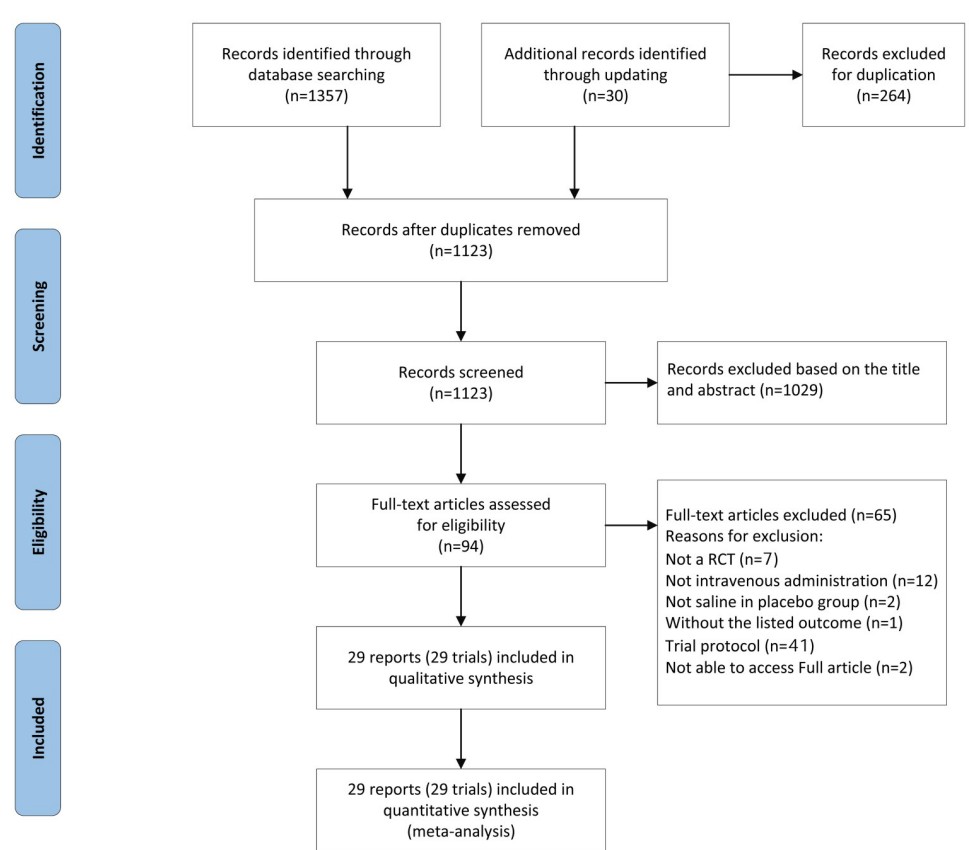

**Fig 1. Literature search and study selection process according to the PRISMA guidelines.**

41, 48]. Twenty studies enrolled both males and females, with the proportions of male patients ranged from 48% to 81% [15, 23, 27–35, 38–43, 45, 47, 48]; two studies enrolled only males [17, 26] and four studies only females [24, 25, 44, 46]; the other three studies did not provide information regarding sex proportion [14, 36, 37]. Anesthetic methods included general anesthesia (n = 23) [15, 23–28, 30–32, 34–36, 38–45, 47, 48], neuraxial anesthesia (n = 2) [17, 46], and combined peripheral nerve/neuraxial block-general anesthesia (n = 4) [14, 29, 33, 37]. The timing of dexmedetomidine administration included intraoperative (n = 14) [17, 23, 25, 27, 29, 31–34, 37, 38, 42–44], postoperative (n = 11) [14, 15, 24, 26, 28, 30, 36, 39–41, 48], and intra- and postoperative (n = 4) [35, 45–47]. The placebo group in all studies received 0.9% saline. The methods used to assess sleep quality included subjective scales (n = 25) [14, 15, 23–27, 29–41, 44–48] and objective parameters (n = 9) [15, 17, 24, 26, 28, 41–43, 48]. Other data extracted from the included studies are provided in S2 Table.

## 3.3 Quality evaluation of included studies

Thirteen studies did not describe the randomization process [15, 23, 24, 26–28, 33, 36–39, 45, 47] and one study did not conduct allocation concealment [46]; two studies had deviations from intended intervention [33, 42] while six studies did not clearly describe if they had deviation from intended intervention [17, 27, 34, 41, 43, 45]; one study had a drop-out rate >20% (31% [20/64]) [17]; one study had high risk of bias in outcome measurements [24] and four studies did not clearly describe if outcome assessors were blinded [27, 29, 33, 38]; 15 studies

**Table 1. Summary of details of the included studies.**

| Author, Year | Sample size (D/C) | Age (D/C) | Male (D/C) | Type of surgery | Type of anesthesia | Postoperative analgesic | Timing of intervention | Strategy of DEX administration | Sleep quality assessment |
|---|---|---|---|---|---|---|---|---|---|
| Chen C, 2016 [23] | 30/30 | 57/60 | 14/15, 48% | Laparoscopic colorectal surgery | GA | PCIA with sufentanil | Intraoperative | 1 μg·kg$^{-1}$ over 10 min, then 0.3 μg·kg$^{-1}$·h$^{-1}$ during surgery | NRS |
| Chen Z, 2017 [24] | 30/29 | 43/45 | 0/0, 0% | Abdominal hysterectomy | GA | PCIA with sufentanil or sufentanil + DEX | Postoperative | PCIA: 0.05 μg·kg$^{-1}$·h$^{-1}$, bolus 0.05 μg·kg$^{-1}$, lockout 10 min | PSG, NRS |
| Dong YS, 2024 [25] | 94/94 | 54/53 | 0/0, 0% | Radical surgery for breast cancer | GA | I.V. flubiprofen or IM tramadol | Intraoperative | 1 μg·kg$^{-1}$ for 15 min, then 0.4 μg·kg$^{-1}$·h$^{-1}$ until the surgical drain started to be placed | AIS |
| Huang J, 2023 [26] | 20/20 | 66/66 | 20/20,100% | Total/partial laryngectomy | GA | PCIA with sufentanil +/- I.V. hydromorphone | Postoperative | 0.3 μg kg$^{-1}$h$^{-1}$ from 9 pm on surgery day to 6 am the next morning | NRS, PSG |
| Jiang Z, 2018 [28] | 65/32 | 64/65 | 40/17, 59% | Laparoscopic gastrointestinal surgery | GA | PCIA with oxycodone or oxycodone + DEX | Postoperative | PCIA: 0.072 μg·kg$^{-1}$·h$^{-1}$ or 0.144 μg·kg$^{-1}$·h$^{-1}$, bolus 0.024 μg·kg$^{-1}$ or 0.048 μg·kg$^{-1}$, lockout 12 min | PSG |
| Kang RA, 2019 [29] | 22/22 | 47/46 | 13/15, 64% | Arthroscopic shoulder surgery | GA + ISBPB | PCIA with fentanyl | Intraoperative | 1 μg·kg$^{-1}$ over 30 min | Likert scale |
| Li HJ, 2018 [30] | 28/29 | 69/67 | 17/15, 56% | Open abdominal surgery | GA | PCIA with morphine or morphine + DEX | Postoperative | PCIA: 2 μg·h$^{-1}$, bolus 4 μg, lockout 8 min | NRS |
| Li S, 2023 [31] | 130/130 | 45/45 | 72/64, 52% | Craniotomy | GA | PCIA with sufentanil | Intraoperative | 0.6 μg·kg$^{-1}$ over 10 min, then 0.4 μg·kg$^{-1}$·h$^{-1}$ until the start of dural closure | RCSQ |
| Liu T, 2022 [32] | 60/60 | 71/72 | 31/29, 50% | Oral and maxillofacial surgery | GA | PCIA with sufentanil | Intraoperative | 0.5 μg·kg$^{-1}$ over 10 min, then 0.4 μg·kg$^{-1}$·h$^{-1}$ until 30 min before end of surgery | RCSQ |
| Liu X, 2020 [33] | 38/37 | 67/70 | 25/26, 68% | Total hip arthroplasty | GA + FICB | PCIA with bupivacaine + sufentanil | Intraoperative | 0.6 μg·kg$^{-1}$·h$^{-1}$ until 30 min before end of surgery | PSQI |
| Lu Y, 2021 [34] | 344/331 | 70/70 | 222/223, 66% | Abdominal surgery | GA | PCIA with sufentanil + flurbiprofen axetil | Intraoperative | 0.5 μg·kg$^{-1}$ over 15 min, then 0.2 μg·kg$^{-1}$·h$^{-1}$ until 30 min before end of surgery | NRS |
| Mao Y, 2020 [35] | 29/29 | 65/63 | 23/24, 81% | Thoracotomy for esophageal cancer | GA | PCIA with sufentanil + flurbiprofen axetil | Intra-and postoperative | 0.5 μg·kg$^{-1}$ over 10 min, 0.2–0.4 μg·kg$^{-1}$·h$^{-1}$ until end of surgery, 0.06 μg·kg$^{-1}$·h$^{-1}$ for 5 days | St.Mary's Hospital Sleep Questionnaire |
| Qin M, 2017 [36] | 29/29 | 59/58 | - - - | Partial laryngectomy | GA | PCIA with sufentanil or sufentanil + DEX | Postoperative | PCIA: 6 μg·h$^{-1}$, bolus 6 μg, lockout 10 min | Complaint of sleep disturbance |
| Shi H, 2020 [37] | 53/53 | 69/69 | - - - | Thoracoscopic lobectomy | GA + thoracic epidural | PCEA with ropivacaine + sufentanil | Intraoperative | 0.5 μg·kg$^{-1}$·h$^{-1}$ until end of surgery | NRS |
| Shi J, 2022 [38] | 142/143 | 58/57 | 79/75, 54% | Craniotomy for HICH | GA | - - - | Intraoperative | 0.5 μg·kg$^{-1}$ within 10 min, 0.5 μg·kg$^{-1}$·h$^{-1}$ until 15 min before end of surgery | PSQI |
| Sui X, 2022 [39] | 140/70 | 60/60 | 98/38, 65% | Colorectal cancer surgery | GA | PCIA with sufentanil or sufentanil + DEX | Postoperative | PCIA: 2.67 μg·h$^{-1}$ or 5.33 μg·h$^{-1}$, bolus 2 μg or 4 μg, lockout 15 min | AIS |
| Sun Y, 2019 [40] | 281/276 | 68/69 | 161/154, 57% | Major noncardiac surgery | GA | PCIA with sufentanil or sufentanil + DEX | Postoperative | PCIA: 0.1 μg·kg$^{-1}$·h$^{-1}$ for 48 h | RCSQ |
| Sun YM, 2022 [41] | 33/35 | 71/66 | 20/20, 59% | Noncardiac surgery | GA | PCIA/PCEA +/- I. V. morphine | Postoperative | 0.1–0.2 μg·kg$^{-1}$·h$^{-1}$ during MV, up to 3 days | RCSQ, PSG |

(*Continued*)

**Table 1.** (Continued)

| Author, Year | Sample size (D/C) | Age (D/C) | Male (D/C) | Type of surgery | Type of anesthesia | Postoperative analgesic | Timing of intervention | Strategy of DEX administration | Sleep quality assessment |
|---|---|---|---|---|---|---|---|---|---|
| Su X, 2016 [14] | 350/350 | ≥65y | - - - | Non-cardiac surgery | GA/GA + epidural | PCIA/PCEA +/-I.V. morphine | Postoperative | 0.1 µg·kg⁻¹·h⁻¹ until 8 am the next morning | NRS |
| Tan W, 2016 [17] | 22/22 | 70/71 | 22/22, 100% | TURP | CSEA | Epidural morphine | Intraoperative | 6 µg·kg⁻¹·h⁻¹ for 10 min, then 1.2 µg·kg⁻¹·h⁻¹ till end of surgery | BIS |
| Tan W, 2016 (2) [42] | 55/53 | 56/56 | 31/27, 54% | Thoracotomy for lung surgery | GA | PCIA with sufentanil | Intraoperative | 1 µg·kg⁻¹ over 10 min | BIS |
| Ting H, 2019 [27] | 173/173 | 70/71 | 89/94, 53% | Radical lung resection | GA | - - - | Intraoperative | 0.5 µg·kg⁻¹ over 20 min, 0.1 µg·kg⁻¹·h⁻¹ until 30 min before end of surgery | PSQI |
| Wu XH, 2016 [15] | 38/38 | 74/76 | 24/20, 58% | Noncardiac surgery | GA | PCIA/PCEA +/- I.V. morphine | Postoperative | 0.1 µg·kg⁻¹·h⁻¹ until 8 am the next morning | NRS, PSG |
| Wu Y, 2022 [43] | 48/48 | 45/42 | 24/29, 55% | Endoscopic sinus surgery | GA | - - - | Intraoperative | 0.5 µg·kg⁻¹ over 10 min, then 0.2 µg·kg⁻¹·h⁻¹ until 30 min before end of surgery | PSG |
| Xu S, 2023 [44] | 80/80 | 50/50 | 0/0, 0% | Laparoscopic hysterectomy | GA | PCIA with sufentanil | Intraoperative | 0.5 µg·kg⁻¹ over 10 min, then 0.4 µg·kg⁻¹·h⁻¹ until 30 min before end of surgery | NRS |
| Yang X, 2015 [45] | 39/40 | 50/51 | 21/21, 53% | Maxillofacial surgery | GA | PCIA with sufentanil | Intra- and postoperative | 0.5 µg·kg⁻¹·h⁻¹ until end of surgery, then 0.2–0.7 µg·kg⁻¹·h⁻¹ until next 6 am | NRS |
| Yu HY, 2019 [46] | 281/276 | 32/31 | 0/0, 0% | Caesarean section | SA | PCIA with sufentanil or sufentanil + DEX | Intra- and postoperative | 0.5 µg·kg⁻¹ over 20 min, then PCIA: 0.04 µg·kg⁻¹·h⁻¹, bolus 2 µg, lockout 8 min | ISI |
| Yu Y, 2023 [47] | 156/154 | 41/39 | 87/92, 58% | Emergency trauma surgery | GA | PCIA with sufentanil +/- IV flurbiprofen axetil | Intra- and postoperative | 0.1 µg·kg⁻¹·h⁻¹ until end of surgery, then 0.1 µg·kg⁻¹·h⁻¹ from 9 pm to 7 am on days 1 to 3 after surgery | NRS |
| Zhang ZF, 2022 [48] | 58/59 | 69/68 | 30/34, 55% | Major noncardiac surgery | GA | PCIA with morphine or morphine + DEX | Postoperative | PCIA: 1.25 µg·h⁻¹, bolus 2.5 µg, lockout 8 min | NRS, PSG |

Abbreviations: D/DEX, dexmedetomidine; C, control; GA, general anesthesia; PCIA, patient controlled intravenous analgesia; NRS, numerical rating scale; PSG, polysomnography; I.V., intravenous; IM, intramuscularly; AIS, Athens insomnia scale; ISBPB, interscalene brachial plexus block; RCSQ, Richards-Campbell sleep questionnaire; FICB, fascia iliaca compartment block; PSQI, Pittsburgh sleep quality index; PCEA, patient controlled epidural analgesia; HICH, hypertensive cerebral hemorrhage; MV, mechanical ventilation; TURP, transurethral resection prostate; CSEA, combined spinal-epidural anesthesia; BIS, bispectral index; SA, spinal anesthesia; ISI, Insomnia severity index.

"- - -" indicate data were not reported.

had unclear risk of reporting bias [24, 25, 27–30, 32–35, 37–39, 43, 47]. The remaining studies had low risks of performance and measurement bias (Fig 2; S3 Table).

## 3.4 Outcome measures

**3.4.1 Primary outcome.** A total of 22 studies reported subjective sleep quality on the first postoperative night [14, 15, 23–32, 34, 37–41, 44, 45, 47, 48]. The result of meta-analysis showed that the subjective sleep score was significantly lower (better) in the dexmedetomidine group than in the placebo group (SMD = -0.8 point, 95% CI -1.1 to -0.6 point, p<0.00001; I² =

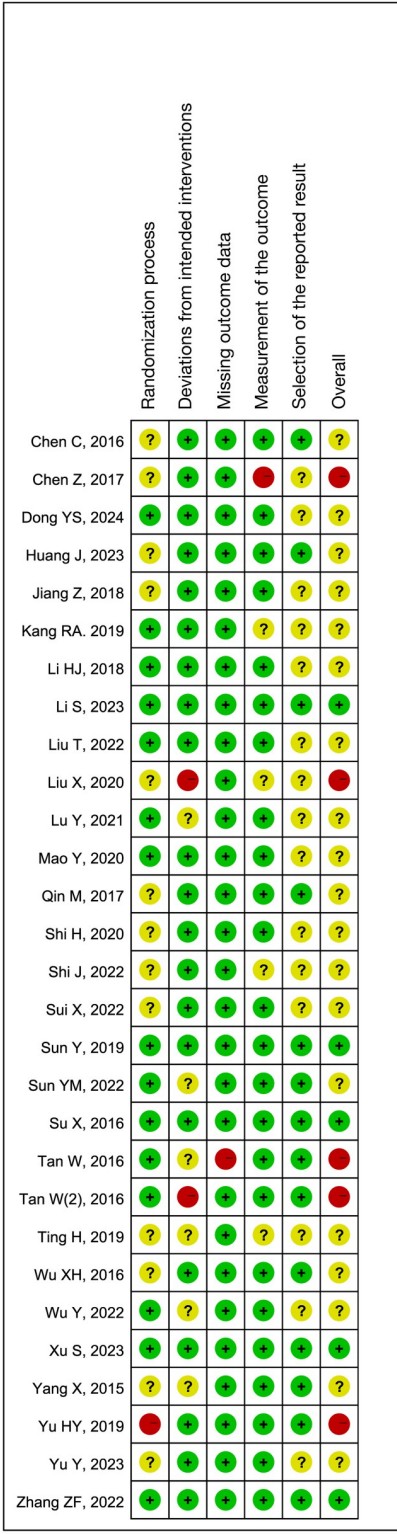

**Fig 2. Risk of bias of the included studies.**

**Table 2. Meta-analyses of primary, exploratory, and safety outcomes.**

| Outcomes | No. of studies | No. of patients | | Heterogeneity (I²) | Pooled MD or SMD or RR (95% CI) | Significance (p) |
|---|---|---|---|---|---|---|
| | | DEX | Control | | | |
| **Primary outcome** | | | | | | |
| Subjective sleep score on the 1st night (point) [14, 15, 23–32, 34, 37–41, 44, 45, 47, 48] | 22 | 2365 | 2246 | 93% | SMD = -0.8 (-1.1, -0.6) | **<0.00001** |
| **Exploratory outcomes** | | | | | | |
| Polysomnographic parameters | | | | | | |
| Sleep efficiency index (SEI; %) [15, 24, 26, 28, 41, 43, 48] | 7 | 268 | 233 | 98% | MD = 12.4 (2.0, 22.9) | **0.02** |
| Arousal index (AI; times/h) [24, 26, 28, 41, 43, 48] | 6 | 237 | 203 | 94% | MD = -2.7 (-4.6, -0.8) | **0.005** |
| Stage 1 of non-REM sleep (N1; %) [15, 24, 26, 28, 41, 48] | 6 | 220 | 185 | 97% | MD = -12.0 (-19.8, -4.2) | **0.003** |
| Stage 2 of non-REM sleep (N2; %) [15, 24, 26, 28, 41, 48] | 6 | 220 | 185 | 95% | MD = 14.1 (6.3, 22.0) | **0.0004** |
| Stage 3 of non-REM sleep (N3; %) [24, 28] | 2 | 95 | 61 | 90% | MD = -0.3 (-3.5, 2.8) | 0.84 |
| REM sleep (%) [24, 26, 28, 41, 43] | 5 | 193 | 162 | 91% | MD = -0.7 (-1.9, 0.5) | 0.25 |
| Subjective sleep score on the 2nd night (point) [14, 15, 24, 27, 28, 30, 31, 34, 38–40, 46–48] | 14 | 2208 | 2083 | 93% | SMD = -0.7 (-0.9, -0.4) | **<0.00001** |
| Subjective sleep score on the 3rd night (point) [14, 15, 25, 27, 31, 34, 39, 40, 47, 48] | 10 | 1756 | 1667 | 90% | SMD = -0.3 (-0.5, -0.1) | **0.009** |
| Subjective sleep score at one week or later (point) [25, 33, 34, 38, 39, 41, 46] | 7 | 1079 | 991 | 95% | SMD = -0.3 (-0.7, 0.1) | 0.14 |
| Numeric rating scale of pain at 24 h (point) [14, 23, 24, 26, 27, 30–37, 39, 40, 42–48] | 22 | 2452 | 2356 | 96% | MD = -0.5 (-0.7, -0.3) | **<0.00001** |
| Morphine equivalent within 7 days (mg) [14, 15, 23, 24, 26, 29, 30, 35, 36, 39, 45, 48] | 12 | 813 | 745 | 98% | MD = -6.1 (-9.1, -3.2) | **<0.0001** |
| Incidence of delirium after surgery (%) [14, 15, 30, 31, 34, 35, 37, 40, 41, 43, 45, 47, 48] | 13 | 1594 | 1577 | 34% | RR = 0.66 (0.50, 0.85) | **0.002** |
| **Safety outcomes** | | | | | | |
| Incidence of bradycardia (%) [14, 15, 28, 31, 32, 37, 38, 40, 41, 44, 45, 47] | 12 | 1427 | 1391 | 40% | RR = 1.72 (1.29, 2.28) | **0.0002** |
| Incidence of hypotension (%) [14, 15, 28, 30–32, 37, 38, 40, 41, 44, 45, 47, 48] | 14 | 1513 | 1479 | 6% | RR = 1.28 (1.04, 1.59) | **0.02** |

Abbreviations: No., number; DEX, dexmedetomidine; MD, mean difference; SMD, standardized mean difference; RR, relative risk; CI, confidence interval; SEI, sleep efficiency index; AI, arousal index; REM, rapid eye movement. P values in bold indicate <0.05.

93%; 22 trials; n = 4611; Table 2). Subgroup analysis according to age showed similar results between patients aged ≥65 years and those <65 years (Fig 3A). Subgroup analysis according to timing of dexmedetomidine administration showed that subjective sleep quality was significantly improved in patients with intra- or postoperative intervention, but not in those with intra- and postoperative intervention. However, the latter phenomenon can be attributed to the small sample size (n = 389) in the two studies testing intra- and postoperative dexmedetomidine [45, 47] (Fig 3B). Asymmetry observed in the funnel plot (S1 Fig in S3 File) showed the existence of publication bias, which were also confirmed by egger test (P = 0.043). Sensitivity analysis showed that the overall conclusion was not changed after excluding eight studies (Table 3) [24, 25, 28–30, 41, 44, 45]. The heterogeneity of conclusion was high (I² = 93%). The quality of evidence was very low according to the GRADE system (S4 Table).

**3.4.2 Exploratory outcomes.** Among PSG parameters on the first night after surgery, SEI (pooled MD = 12.4%, 95% CI 2.0 to 22.9%, p = 0.02; I² = 98%; 7 trials; n = 501; S2A Fig in S3 File) [15, 24, 26, 28, 41, 43, 48] and %N2 sleep (pooled MD = 14.1%, 95% CI 6.3 to 22.0%, p = 0.0004; I² = 95%; 6 trials; n = 405; S3B Fig in S3 File) [15, 24, 26, 28, 41, 48] were

(A)

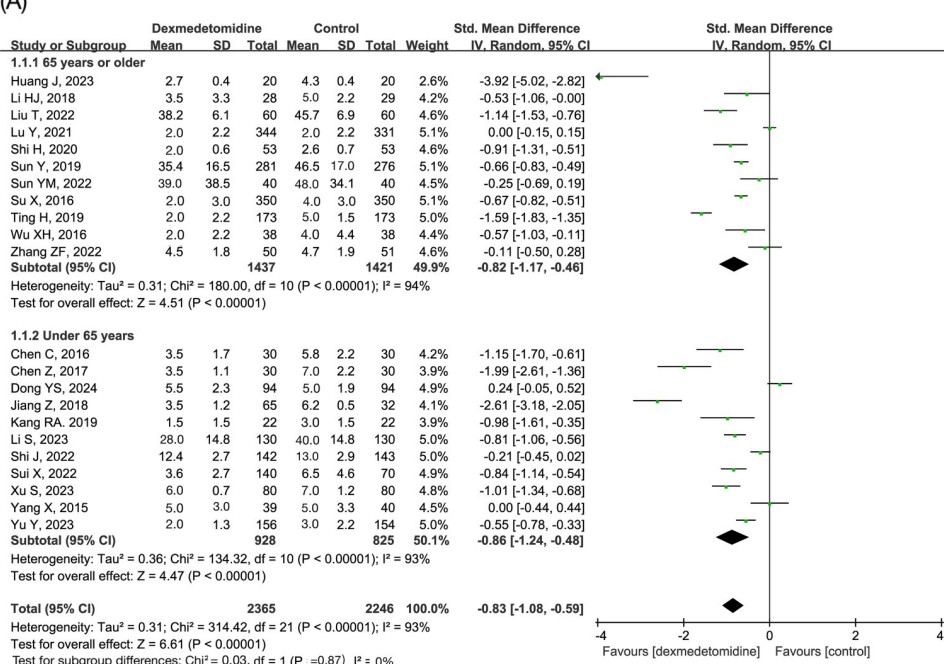

(B)

**Fig 3. Forest plot of the pooled analysis showing subjective sleep score on the first night after surgery.** Subgroup analyses were performed according to age (A) and timing of dexmedetomidine administration (B).

**Table 3. Sensitivity analyses after excluding trials that had small sample sizes and reported medians, or were judged at high risk of bias.**

| Outcomes | No. of studies | No. of patients | | Heterogeneity ($I^2$) | Pooled MD or SMD or RR (95% CI) | Significance (p) | Test for subgroup difference (p) |
| --- | --- | --- | --- | --- | --- | --- | --- |
| | | DEX | Control | | | | |
| **Primary outcome** | | | | | | | |
| Subjective sleep score on the 1st night (point) [14, 15, 23, 26, 27, 31, 32, 34, 37–40, 47, 48] | 14 | 1967 | 1879 | 93% | SMD = -0.8 (-1.1, -0.5) | <**0.00001** | - - - |
| Stratified according to age | | | | | | | |
| ≥65 y [14, 15, 26, 27, 32, 34, 37, 40, 48] | 9 | 1369 | 1352 | 95% | SMD = -0.9 (-1.3, -0.5) | <**0.00001** | 0.33 |
| <65 y [23, 31, 38, 39, 47] | 5 | 598 | 527 | 80% | SMD = -0.7 (-1.0, -0.4) | <**0.00001** | |
| Stratified according to administration time | | | | | | | |
| Intraoperative [23, 27, 31, 32, 34, 37, 38] | 7 | 932 | 920 | 96% | SMD = -0.8 (-1.3, -0.3) | **0.002** | 0.39 |
| Postoperative [14, 15, 26, 39, 40, 48] | 6 | 879 | 805 | 88% | SMD = -0.8 (-1.2, -0.5) | <**0.00001** | |
| Intra- and postoperative [47] | 1 | 156 | 154 | - - - | SMD = -0.6 (-0.8, -0.3) | <**0.00001** | |
| **Exploratory outcomes** | | | | | | | |
| Polysomnographic parameters | | | | | | | |
| Sleep efficiency index (SEI; %) [26, 43] | 2 | 65 | 66 | - - - | - - - | - - - | - - - |
| Arousal index (AI; times/h) [26, 43] | 2 | 65 | 66 | - - - | - - - | - - - | - - - |
| Stage 1 of non-REM sleep (N1; %) [26] | 1 | 17 | 18 | - - - | - - - | - - - | - - - |
| Stage 2 of non-REM sleep (N2; %) [26] | 1 | 17 | 18 | - - - | - - - | - - - | - - - |
| Stage 3 of non-REM sleep (N3; %) | 0 | 0 | 0 | - - - | - - - | - - - | - - - |
| REM sleep (%) [26, 43] | 2 | 65 | 66 | - - - | - - - | - - - | - - - |
| Subjective sleep score on the 2nd night (point) [14, 15, 27, 31, 34, 38–40, 47, 48] | 10 | 1804 | 1716 | 90% | SMD = -0.4 (-0.7, -0.2) | <**0.0002** | - - - |
| Subjective sleep score on the 3rd night (point) [14, 15, 27, 31, 34, 39, 40, 47, 48] | 9 | 1662 | 1573 | 91% | SMD = -0.3 (-0.6, -0.1) | **0.006** | - - - |
| Subjective sleep score at one week or later (point) [34, 38, 39] | 3 | 626 | 544 | 97% | SMD = -0.6 (-1.4, 0.2) | 0.11 | - - - |
| Numeric rating scale of pain at 24 h (point) [14, 23, 26, 31, 32, 37, 40, 43, 47] | 9 | 1128 | 1121 | 93% | MD = -0.4 (-0.6, -0.2) | **0.0001** | - - - |
| Morphine equivalent within 7 days (mg) [14, 23, 26, 29, 35, 36, 39, 45] | 8 | 659 | 590 | 98% | MD = -5.6 (-8.9, -2.3) | **0.001** | - - - |
| Incidence of delirium after surgery (%) [14, 15, 30, 31, 34, 35, 37, 40, 41, 43, 45, 47, 48] | 13 | 1594 | 1577 | 34% | RR = 0.66 (0.50, 0.85) | **0.002** | - - - |
| Safety outcomes | | | | | | | |
| Incidence of bradycardia (%) [14, 15, 28, 31, 32, 37, 38, 40, 41, 44, 45, 47] | 12 | 1427 | 1391 | 40% | RR = 1.72 (1.29, 2.28) | **0.0002** | - - - |
| Incidence of hypotension (%) [14, 15, 28, 30–32, 37, 38, 40, 41, 44, 45, 47, 48] | 14 | 1513 | 1479 | 6% | RR = 1.28 (1.04, 1.59) | **0.02** | - - - |

Abbreviations: No., number; DEX, dexmedetomidine; MD, mean difference; SMD, standardized mean difference; RR, relative risk; CI, confidence interval; SEI, sleep efficiency index; AI, arousal index; REM, rapid eye movement. P values in bold indicate <0.05.

significantly higher (better), and AI (pooled MD = -2.7 times/h, 95% CI -4.6 to -0.8 times/h, p = 0.005; $I^2$ = 94%; 6 trials; n = 440; S2B Fig in S3 File) [24, 26, 28, 41, 43, 48] and %N1 sleep (pooled MD = -12.0%, 95% CI -19.8 to -4.2%, p = 0.003; $I^2$ = 97%; 6 trials; n = 405; S3A Fig in S3 File) [15, 24, 26, 28, 41, 48] were significantly lower (better) in the dexmedetomidine group than in the placebo group (Table 2); the quality of evidence was low (S4 Table). %N3 and % REM sleep did not differ significantly between the two groups (Table 2; S3C and S3D Fig in S3 File); the quality of evidence was very low (S4 Table). After excluding studies that had small sample sizes and reported medians or at high risk of bias, only two studies existed and therefore we did not conduct sensitivity analysis for PSG parameters (Table 3).

Subjective sleep score on the second night after surgery was significantly lower (better) in the dexmedetomidine group than in the placebo group (SMD = -0.7 point, 95% CI -0.9 to -0.4 point, p<0.00001; $I^2$ = 93%; 14 trials; n = 4291; Table 2; S4 Fig in S3 File) [14, 15, 24, 27, 28, 30, 31, 34, 38–40, 46–48]. The heterogeneity of conclusions was high ($I^2$ = 93%). Sensitivity analysis showed that the conclusion was not changed after excluding four studies (Table 3) [24, 28, 30, 46]. The quality of evidence was very low (S4 Table).

Subjective sleep score on the third night after surgery was also significantly lower (better) in the dexmedetomidine group than in the placebo group (SMD = -0.3 point, 95% CI -0.5 to -0.1 point, p = 0.009; $I^2$ = 90%; 10 trials; n = 3423; Table 2; S5 Fig in S3 File) [14, 15, 25, 27, 31, 34, 39, 40, 47, 48]. The heterogeneity of conclusion was high ($I^2$ = 90%). Sensitivity analysis showed that conclusion was not changed after excluding one study (Table 3) [25]. The quality of evidence was low (S4 Table).

Subjective sleep score at one week or later after surgery did not differ significantly between the two groups (Table 2; S6 Fig in S3 File) [25, 33, 34, 38, 39, 41, 46]. The heterogeneity of conclusion was high ($I^2$ = 95%). Sensitivity analysis showed that the conclusion was not changed after excluding four studies (Table 3) [25, 33, 41, 46]. The quality of evidence was very low (S4 Table).

Numeric rating scale of pain at 24 hours after surgery (MD = -0.5 point, 95% CI -0.7 to -0.3 point, p<0.00001; $I^2$ = 96%; 22 trials; n = 4808; S7 Fig in S3 File) [14, 23, 24, 26, 27, 30–37, 39, 40, 42–48], morphine equivalent consumption within 7 days (MD = -6.1 mg, 95% CI -9.1 to -3.2 mg, p<0.0001; $I^2$ = 98%; 12 trials; n = 1558; S8 Fig in S3 File) [14, 15, 23, 24, 26, 29, 30, 35, 36, 39, 45, 48], and incidence of delirium after surgery (RR = 0.66, 95% CI 0.50 to 0.85, p = 0.002; $I^2$ = 34%; 13 trials; n = 3171; S9 Fig in S3 File) [14, 15, 30, 31, 34, 35, 37, 40, 41, 43, 45, 47, 48] were also significantly lower in the dexmedetomidine group than in the placebo group (Table 2). Sensitivity analyses showed that conclusions were not changed (Table 3). The quality of evidence was low for numeric rating scale of pain at 24 h and morphine equivalent consumption within 7 days, and was moderate for incidence of delirium after surgery (S4 Table).

**3.4.3 Safety outcomes.** The incidences of bradycardia (RR = 1.72, 95% CI 1.29 to 2.28, p = 0.0002; $I^2$ = 40%; 12 trials; n = 2818; Table 2; S10A Fig in S3 File) [14, 15, 28, 31, 32, 37, 38, 40, 41, 44, 45, 47] and hypotension (RR = 1.28, 95% CI 1.04 to 1.59, p = 0.02; $I^2$ = 6%; 14 trials; n = 2992; Table 2; S10B Fig in S3 File) [14, 15, 28, 30–32, 37, 38, 40, 41, 44, 45, 47, 48] was significantly higher in the dexmedetomidine group. The heterogeneities of conclusions were low ($I^2$ = 40% and 6%, respectively). Sensitivity analyses showed that incidences of bradycardia and hypotension were not changed (Table 3). The quality of evidence was moderate for both bradycardia and hypotension (S4 Table).

## 4. Discussion

Results of our meta-analysis showed that perioperative dexmedetomidine administered intravenously improved subjective sleep quality on the first night after surgery. In accord with these, results of PSG monitoring showed that perioperative dexmedetomidine improved sleep structure on the first night after surgery, as manifested by increased SEI and %N2 sleep and decreased AI and %N1 sleep. Perioperative dexmedetomidine also improved subjective sleep quality on the second and third nights after surgery. However, heterogeneities were high, and qualities of evidence were low or very low.

Increasing studies investigated the effect of dexmedetomidine on postoperative sleep [15, 41, 43, 48]. However, participants and interventional strategies are various in these studies. Consequently, whether perioperative dexmedetomidine administration can improve sleep

quality, and whether the sleep-promoting effect of dexmedetomidine is related to age of participants or timing of administration remains inconclusive. Several meta-analyses evaluated the effect of dexmedetomidine on postoperative delirium [49–53], but only a few summarized its effect on postoperative sleep, including a systematic review and qualitative summary of overall subjective and objective sleep quality [18] and a meta-analysis regarding PSG parameters [19]. Our systematic review and meta-analysis quantitatively evaluated the effects of perioperative intravenous dexmedetomidine on sleep quality and safety outcomes.

Among studies included in this systematic review, subjective sleep quality was mainly assessed with the numerical rating scale [14, 15, 23, 24, 26, 30, 34, 37, 44, 45, 47, 48], followed by Richards-Campbell sleep questionnaire [31, 32, 40, 41] and Pittsburgh sleep quality index [23, 27, 33, 38]; a few studies used Athens Insomnia Scale [25, 39], Insomnia Severity Index [46], Likert Scale [29], and St. Mary's Hospital Sleep Questionnaire [35]. Objective sleep quality was mostly monitored with PSG [15, 24, 26, 28, 41, 43, 48], which is the gold standard for sleep structure evaluation, and also with Bispectral Index (BIS) [17, 42]. Since studies assessed subjective sleep quality with different scales, we therefore adopted SMD and 95% CI as the effect measure according to the Cochrane Handbook [21]. We found that perioperative dexmedetomidine improved subjective sleep quality for up to three days after surgery. The improvement of subjective sleep quality was also confirmed by the PSG parameters which showed increased sleep efficiency index and %N2 sleep and decreased arousal index and %N1 sleep on the first night after surgery. For skewed data, mean (and SD) converted from median (and interquartile range) do not accurately reflect distribution of results. In sensitivity analysis, we excluded studies with small sample sizes and outcomes expressed in medians or those at high risk of bias. Results of sensitivity analysis did not change our primary conclusion.

Unlike traditional sedative drugs, dexmedetomidine produces sedative effects by activating the endogenous sleep-promoting pathway [54], producing a state resembling non-rapid eye movement sleep [55, 56]. Indeed, studies showed that night-time infusion of sedative-dose dexmedetomidine (0.6 µg/kg/h) improved sleep quality in ICU patients with mechanical ventilation by decreasing sleep fragmentation and %N1 sleep, increasing sleep efficiency and %N2 sleep, and shifting sleep to the night [57]. Night-time infusion of low-dose dexmedetomidine (0.1 µg/kg/h) improved sleep quality of ICU patients without mechanical ventilation in a similar way [15]. In a recent trial, mini-dose dexmedetomidine (0.02 µg/kg/h) as a supplement to patient-controlled analgesia also improved sleep quality as above without producing sedation in non-ICU patients [48]. Our current results are in line with most of the available studies on subjective feelings and objective parameters of sleep [18, 19].

Surgery-related pain and inflammation, as well as concomitant opioid use, are important causes of postoperative sleep disturbances [6, 58, 59]. In accord with others [60–62], our results showed that perioperative dexmedetomidine significantly improved analgesia and reduced opioid consumption. In previous studies, dexmedetomidine as an anesthesia adjuvant attenuated perioperative stress response and inflammation [63]. These effects might also have contributed to the sleep-promoting effects of dexmedetomidine. As can be expected, perioperative dexmedetomidine decreased the occurrence of postoperative delirium; similar results have been reported by others [64].

Older patients are more likely to develop sleep disturbances after surgery [59, 65, 66], and those who developed sleep disturbances are at increased risk of postoperative delirium [9, 10], delayed recovery [67], and even worse long-term quality of life [12]. Results of our subgroup analyses showed that dexmedetomidine was equally effective in improving sleep quality in patients aged under 65 years and those aged 65 years or older. In our results, both intra- and postoperative dexmedetomidine administration improved sleep quality on the first night after

surgery. Dexmedetomidine can thus be administered in older and non-older patients and intraoperatively and/or postoperatively.

Side effects of dexmedetomidine are mainly hemodynamic and include bradycardia and hypotension [68]. We also found that bradycardia and hypotension were more common in patients given dexmedetomidine. Our results together with others suggest that careful monitoring is necessary during dexmedetomidine administration, especially in older patients.

There are some notable limitations of this review. First, heterogeneities were high in most of our outcome measures. Of the included studies, participants, surgical types, interventional strategies, and sleep quality assessment tools were highly variable; funnel plot and egger test also showed the existence of publication bias. All these might have contributed to heterogeneity generation. Second, we excluded trials that compared dexmedetomidine with other sedative drugs such as propofol. We do not know if other sedatives were equally effective in improving postoperative sleep. Third, the doses of dexmedetomidine were different in our included trials and were sometimes adjusted according to the patients' conditions. We do not know the optimal dose or infusion rate of dexmedetomidine for sleep promotion. Fourth, most of included trials had small sample sizes and unclear risk of bias; the resulting evidence were of low or very low qualities. Therefore, our results should be interpreted with caution. The observed effects need to be confirmed in future large sample size trials.

## 5. Conclusions

In summary, among patients who underwent noncardiac surgery, perioperative dexmedetomidine via intravenous administration may improve subjective sleep quality early after surgery; it may also improve sleep structure by increasing sleep efficiency and %N2 sleep and decreasing arousal index and %N1 sleep. Use of dexmedetomidine increases bradycardia and hypotension. Further studies are required to confirm our results, to explore the optimal dose/ infusion rate of dexmedetomidine for sleep promotion, and to clarify the effects of sleep promotion on clinical outcomes.

## Supporting information

**S1 File. PRISMA 2020 checklist.**
(DOCX)

**S2 File. Search strategy.**
(DOCX)

**S3 File. Funnel plot for subjective sleep score on the first night after surgery and forest plots of the pooled analyses showing exploratory and safety outcomes.**
(PDF)

**S1 Table. All studies identified in the literature search.**
(DOCX)

**S2 Table. Summary of details of the included studies.**
(DOCX)

**S3 Table. Risk of bias of the included studies.**
(DOCX)

**S4 Table. GRADE quality of evidence assessment for each outcome.**
(DOCX)

## Acknowledgments

The authors gratefully acknowledge Dr. Shuang-Jie Cao (Department of Anesthesiology, Peking University First Hospital, Beijing, China) for her help in literature search and Dr. Zong-Yang Qu (Department of Anesthesiology, Beijing Hospital, National Center of Gerontology) for his help in study selection and data extraction.

## Author Contributions

**Conceptualization:** Dong-Xin Wang.

**Data curation:** Lin Wang, Xin-Quan Liang.

**Formal analysis:** Lin Wang.

**Funding acquisition:** Dong-Xin Wang.

**Investigation:** Lin Wang.

**Methodology:** Lin Wang, Yan-Xia Sun, Zhen Hua.

**Project administration:** Dong-Xin Wang.

**Supervision:** Zhen Hua, Dong-Xin Wang.

**Validation:** Xin-Quan Liang, Zhen Hua.

**Writing – original draft:** Lin Wang.

**Writing – review & editing:** Yan-Xia Sun, Zhen Hua, Dong-Xin Wang.

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
