## [Editor Report · Decision Letter 0]

4 Sep 2024

PONE-D-24-36740Effect of perioperative dexmedetomidine on sleep quality in adult patients after noncardiac surgery: a systematic review and meta-analysis of randomized trialsPLOS ONE

Dear Dr. Wang,

Thank you for submitting your manuscript to PLOS ONE. After careful consideration, we feel that it has merit but does not fully meet PLOS ONE’s publication criteria as it currently stands. Therefore, we invite you to submit a revised version of the manuscript that addresses the points raised during the review process.

We look forward to receiving your revised manuscript.

Kind regards,

Jiawen Deng

Academic Editor

PLOS ONE

Journal Requirements:

"The study was funded by National Natural Science Foundation of China (No. 82293644; Dong-Xin Wang), National High Level Hospital Clinical Research Funding (High Quality Clinical Research Project of Peking University First Hospital No. 2022CR78; Dong-Xin Wang)."

3. As required by our policy on Data Availability, please ensure your manuscript or supplementary information includes the following: 

**Additional Editor Comments:**

I would like to applaud your submission as it exceeds the quality of many other systematic review submissions that we receive at PLOS ONE. However, there are several deficiencies that I would like you to address before I can send this out for further peer-review.

1) Please provide a clear and concrete rationale for your addition of the secondary outcomes and deviation from your original protocol, backed by literatures or clinical knowledge. It is very, very bad practice to include additional outcomes after data extraction because "these results were also investigated". 2) Some PRISMA 2020 items were checked off but not completed. Confidence of evidence (item 15 and item 22 on PRISMA 2020) is a major one. This will need to be done using the GRADE framework. Please review the GRADE methodology and add a GRADE summary of findings table, and please review the PRISMA checklist for any other items that you may have missed. 3) For imputing mean and SD from other summary and variance metrics, it is usually a good idea to perform tests of normality first. Please look into this and add this analysis. 4) You seem to be using the original risk of bias tool (RoB 1.0) from Cochrane. As you know, Cochrane is now recommending their latest tool, RoB-2, and your methods are out of date. Please re-conduct your risk of bias assessment using the updated RoB framework. Note that your submission will still be subjected to external peer-review upon resubmission.

---

## [Author Response · Author response to Decision Letter 0]

22 Oct 2024

Point-by-point response to editor

Journal Requirements:

Requirement 1. Please ensure that your manuscript meets PLOS ONE's style requirements, including those for file naming. 

Response: We rechecked and revised the manuscript again to ensure that it meets PLOS ONE’s style requirements.

Requirement 2. Please state what role the funders took in the study. If the funders had no role, please state: “The funders had no role in study design, data collection and analysis, decision to publish, or preparation of the manuscript.” If this statement is not correct you must amend it as needed. Please include this amended Role of Funder statement in your cover letter; we will change the online submission form on your behalf.

Response: The funders had no role in the study. We clarified this in the submission system: “Funded by National Natural Science Foundation of China (Major Program No. 82293644) and National High Level Hospital Clinical Research Funding (High Quality Clinical Research Project of Peking University First Hospital No. 2022CR78). The funders had no role in study design, data collection and analysis, decision to publish, or preparation of the manuscript.”

Requirement 3. As required by our policy on Data Availability, please ensure your manuscript or supplementary information includes the following: 

#1. A numbered table of all studies identified in the literature search, including those that were excluded from the analyses. For every excluded study, the table should list the reason(s) for exclusion. If any of the included studies are unpublished, include a link (URL) to the primary source or detailed information about how the content can be accessed. 

Response: In the supplementary information, we provide S1 Table which listed all studies identified in the literature search, including those that were excluded from the analyses. For every excluded study, we listed the reason for exclusion. No unpublished study was included.

#2. A table of all data extracted from the primary research sources for the systematic review and/or meta-analysis. The table must include the following information for each study: Name of data extractors and date of data extraction and confirmation that the study was eligible to be included in the review. All data extracted from each study for the reported systematic review and/or meta-analysis that would be needed to replicate your analyses. If data or supporting information were obtained from another source (e.g. correspondence with the author of the original research article), please provide the source of data and dates on which the data/information were obtained by your research group. 

Response: In the manuscript and supplementary information, we provide Table 1 and S2 Table which listed all data extracted from the primary research sources for the systematic review and meta-analysis, including name of data extractors, date of data extraction, confirmation that the study was eligible to be included in the review, and all data extracted from each study for the reported systematic review and/or meta-analysis. All data were obtained from original article.

#3. If applicable for your analysis, a table showing the completed risk of bias and quality/certainty assessments for each study or outcome. Please ensure this is provided for each domain or parameter assessed. For example, if you used the Cochrane risk-of-bias tool for randomized trials, provide answers to each of the signalling questions for each study. If you used GRADE to assess certainty of evidence, provide judgements about each of the quality of evidence factor. This should be provided for each outcome. 

Response: In the supplementary information, we provide S3 Table which listed the risk of bias of included studies, and S4 Table which listed the quality of evidence assessment for each outcome according to the GRADE system. 

#4. An explanation of how missing data were handled. 

Response: We clarified this in the revised manuscript: “The corresponding authors were contacted for missing data or un-reported information. If there were no responses after three contacts, we estimated data according to figures in the original articles.” (Page 7, lines 109-111).

#5. This information can be included in the main text, supplementary information, or relevant data repository. Please note that providing these underlying data is a requirement for publication in this journal, and if these data are not provided your manuscript might be rejected. 

Response: These data are included in the main text and supplementary information.

Additional Editor Comments:

I would like to applaud your submission as it exceeds the quality of many other systematic review submissions that we receive at PLOS ONE. However, there are several deficiencies that I would like you to address before I can send this out for further peer-review.

Comment 1. Please provide a clear and concrete rationale for your addition of the secondary outcomes and deviation from your original protocol, backed by literatures or clinical knowledge. It is very, very bad practice to include additional outcomes after data extraction because "these results were also investigated".

Response: Thank you for your comments. We realized that adding secondary outcomes after data extraction is not a good practice. In the revised “2.1 Protocol and registration” section, we changed “secondary outcomes” to “exploratory outcomes”: “We added exploratory and safety outcomes after data collection because these results were important to clarify the effects of dexmedetomidine.” (Page 6, lines 81-82). We also changed the corresponding expressions in the manuscript throughout. 

Comment 2. Some PRISMA 2020 items were checked off but not completed. Confidence of evidence (item 15 and item 22 on PRISMA 2020) is a major one. This will need to be done using the GRADE framework. Please review the GRADE methodology and add a GRADE summary of findings table, and please review the PRISMA checklist for any other items that you may have missed.

Response: Thank you very much for reminding us. We reviewed and revised the PRISMA checklist carefully to ensure that we completed all items. We also reviewed the GRADE methodology and added a GRADE summary of findings in S4 Table. 

Comment 3. For imputing mean and SD from other summary and variance metrics, it is usually a good idea to perform tests of normality first. Please look into this and add this analysis.

Response: We appreciate your valuable comments. When included studies reported only median and interquartile range, we contacted the authors to obtain the corresponding mean and standard deviation (SD). If there was no response after three contacts, mean and SD were estimated according to the reported formula. However, the original data were not available because the authors did not respond, so we could not perform tests of normality. We agree that mean (and SD) converted from median (and interquartile range) do not accurately reflect distribution of results for skewed data. We therefore performed sensitivity analysis after excluding studies with small sample sizes and outcomes expressed in medians. Results of sensitivity analysis did not change our primary conclusion (Table 3). 

We also discussed this in the “Discussion” section: “For skewed data, mean (and SD) converted from median (and interquartile range) do not accurately reflect distribution of results. In sensitivity analysis, we excluded studies with small sample sizes and outcomes expressed in medians or those at high risk of bias. Results of sensitivity analysis did not change our primary conclusion.” (Page 24, lines 312-316).

Comment 4. You seem to be using the original risk of bias tool (RoB 1.0) from Cochrane. As you know, Cochrane is now recommending their latest tool, RoB-2, and your methods are out of date. Please re-conduct your risk of bias assessment using the updated RoB framework.

Response: Thank you very much. We re-conducted the risk of bias assessment using the updated RoB framework (RoB-2). Results are presented in Fig 2 and S3 Table.

---

## [Editor Report · Decision Letter 1]

18 Nov 2024

Effect of perioperative dexmedetomidine on sleep quality in adult patients after noncardiac surgery: a systematic review and meta-analysis of randomized trials

PONE-D-24-36740R1

Dear Dr. Wang,

We’re pleased to inform you that your manuscript has been judged scientifically suitable for publication and will be formally accepted for publication once it meets all outstanding technical requirements.

Kind regards,

Jiawen Deng

Academic Editor

PLOS ONE
---

## [Editor Report · Acceptance letter]

22 Nov 2024

PONE-D-24-36740R1 

PLOS ONE

Dear Dr. Wang, 

I'm pleased to inform you that your manuscript has been deemed suitable for publication in PLOS ONE. Congratulations! Your manuscript is now being handed over to our production team.

Kind regards, 

on behalf of

Dr. Jiawen Deng 

Academic Editor

PLOS ONE